# A Spatio-Temporal Analysis of the Ecological Compensation for Cultivated Land in Northeast China

Lu Wang [1], Bonoua Faye [2], Quanfeng Li [1,2,3,*] and Yunkai Li [2]

[1] School of Economics and Management, Northeast Agricultural University, Harbin 150030, China; b230801013@neau.edu.cn
[2] School of Public Administration and Law, Northeast Agricultural University, Harbin 150030, China; bonoua.faye2021@neau.edu.cn (B.F.); a12190433@neau.edu.cn (Y.L.)
[3] Land Remote Sensing Big Data Technology Innovation Center, Harbin 150030, China
[*] Correspondence: quanfeng.li@neau.edu.cn; Tel.: +86-18800430545

**Abstract:** Ecological compensation for cultivated land is a prominent means to coordinate the protection and utilization of cultivated land ecosystems. This study assessed the ecological compensation for cultivated land, considering both the ecological footprint and value of ecosystem services. We used the ecological footprint model to calculate the ecological footprint and ecological carrying capacity of cultivated land, combined with the value of its ecosystem services, with a focus on estimating its ecological compensation standard, and we analyzed the temporal and spatial distribution characteristics of Northeast China. We found that the ecological footprint and ecological carrying capacity of cultivated land showed a fluctuating growth trend in Northeast China from 2000 to 2020, increasing by $288.83 \times 10^5$ ha and $131.37 \times 10^5$ ha, respectively. The spatial distribution of cultivated land's ecological footprint and ecological carrying capacity showed growth from the southwest to the northeast. The value of its ecosystem services presented an overall trend of growth over the past 20 years, increasing by CNY 13.64 billion, or an increase of 12.47%. In terms of spatial distribution, the trends of the ecological compensation for cultivated land showed obvious differences. This study mainly focused on black soil cultivated land, and its results are helpful for governments in different countries solving similar problems in terms of the ecological compensation for cultivated land. This study will provide a valuable reference to measure the compensation standard scientifically and to provide policy recommendations for sustainable cultivated land's protection and utilization.

**Keywords:** cultivated land; ecological footprint model; ecosystem services; ecological compensation; Northeast China

## 1. Introduction

Destructive resource utilization has become a prominent constraint to sustainable development [1,2]. Currently, economic growth is faster than ecological self-recovery, destroying the ecological environment [3]. As such, balancing the link between economic growth and ecological protection has become a significant topic of research [4]. Ecological compensation is a vital means of coordinating the interests of various shareholders [5,6]. It provides a comprehensive management system for ecological protection, restoration, and compensation [7]. Ecological compensation is also essential to achieve sustainable development. Governments and the public are increasingly concerned about ecological compensation [8]. Therefore, ecological compensation is a widespread practice adopted in various ecosystems across nations worldwide [9,10], such as cultivated land [11], grasslands [12], forests [13], and wetlands [14]. In particular, ecological compensation practices in cultivated land have made positive progress [15,16].

Cultivated land-related ecological compensation (CLEC) is a crucial strategy to address the imbalance between agricultural productivity and ecological conservation [17]. It is also used to solve the destruction of cultivated land ecosystems caused by irrational resource

utilization [18]. As an important ecological component, governments and the public have participated in CLEC to protect cultivated land [19]. However, the ecological problems with cultivated land are often not alleviated because the ecosystem's service value is not reflected. Moreover, the ecological benefits of cultivated land have an externality and non-exclusivity. Their marginal private costs or marginal benefits deviate from their marginal social costs. The key to solving this conflict between economic development and ecological environmental protection lies in internalizing the externality, and CLEC is one of the effective ways of doing this. This research uses cultivated land externalities and ecosystem service theory to quantify CLEC.

In recent years, research on the ecological compensation standards for cultivated land has gradually become a hot topic in academia and government. Reasonably defining the standards of CLEC is a core component of protecting the ecosystems in cultivated land [20]. Many scholars have used the ecosystem services value (ESV) method of assessment, the contingent valuation method (CVM), and the opportunity cost approach to calculate the standards for CLEC [21]. The ESV assessment method is an efficient way to address the problem of CLEC. The four categories of the ESV (supply, regulation, culture, and support) cover diverse regions are used to determine the amount of ecological compensation required [22,23]. Many researchers have used this method to calculate the regional ESV [21,24,25], but only some scholars have combined an analysis of the ESV and CLEC. In addition, the ecological compensation standard is established upon the foundation of the ESV [26,27]. The CVM considers individuals' willingness to pay for or accept enhancements in their environmental conditions [28]. Conversely, the opportunity cost approach weighs the expense of forgoing potential opportunities for economic progress to safeguard ecological functions [29]. Moreover, this approach typically computes the standard for compensation based on the direct input costs, rendering it more applicable when one cannot directly quantify the social and economic benefits. Numerous scholars have extensively researched these methods, which have reached a level of relative maturity.

Ecosystem services are one of the core elements of sustainable development research, and it is difficult for a single ecological footprint or ecosystem service value approach to adequately reflect the quality of regional cultivated land's ecosystems [30]. Moreover, the traditional ecological footprint model ignores the diversity of ecosystem functions, making it difficult to track the flow of the ESV [2]. The integration of ecosystem services with ecological footprint modeling still needs further exploration. Currently, many scholars have made functional explorations of the combination of ecosystem services and ecological footprints, which have greatly enriched the results of relevant research [16,31]. Some studies have introduced the value of ecological services into examinations of the balance between the supply and demand of land resources and have carried out assessments of ecological compensation and the carrying capacity of land resources [15].

Meanwhile, some scholars have combined the ecological footprint with the ESV method and have proposed the ecological footprint–service value method [32]. However, these studies have omitted the equivalence factor and yield factor. They only classified and compared, and did not form an overall judgment or realize a unified method of accounting for the regional ecological status. Therefore, this study constructed an ecological footprint model guided by the ESV theory to unify the ecological footprint with the study area.

China's exploration of CLEC began relatively late. The concept of ecological compensation in China is closely aligned with the concept of payment for ecosystem services [33]. CLEC has made significant progress in theoretical research and practical applications in recent years. Nevertheless, previous studies have certain shortcomings. In theory, Chinese scholars have predominantly relied on the value of ecosystem services to establish standards for ecological compensation, often supplementing it with other methodologies to offer scientific backing for the determined CLEC [34]. However, static assessments of the ESV have been the norm, overlooking the variations in natural conditions across different regions. As such, we have used the ecological footprint model, analyzed the spatio-temporal changes, and evaluated the ESV in Northeast China. China has imple-

mented CLEC initiatives in numerous regions under the government's guidance [35,36]. Nonetheless, challenges persist, including residents' limited enthusiasm for participating in processes of development and the sluggishness of cross-regional compensation procedures. These issues primarily stem from ambiguities in defining the entities responsible for providing and receiving CLEC, insufficient compensation, and a need for mechanisms for sharing the benefits of the synergistic development of cultivated land [37]. These factors collectively impede the effective implementation of CLEC.

Northeast China (NC) is a significant food-producing area, responsible for a quarter of the country's grain production [38]. At the time of the current study, protecting the cultivated land's ecological ecosystem in NC is a heated issue. However, due to the continual growth of the population, NC is facing severe environmental pressure, such as the degradation of the ecosystems of cultivated land, nutrient imbalance, and desertification [37]. As such, NC has gradually evolved from a natural ecosystem of forests and grasses to an artificially cultivated land ecosystem [38]. Meanwhile, the region is slowly transitioning from an ecologically functioning area to an ecologically fragile area. Its ecological carrying capacity is progressively diminishing, significantly impacting the sustainability of food production and regional ecological stability [39]. Currently, the government has established measures to protect cultivated land to promote the sustainable use of resources, such as laws on the protection of black soil and conservation tillage strategies [40]. However, the amount of CLEC has rarely been measured in NC. Therefore, an accurate measurement of the CLEC is crucial for protecting black soil in NC.

From these research deficiencies, this research is a dynamic and multifaceted process that involves continuously, exploring new ideas concerning the relationship between land use and ecological compensation from the footprint model. From a theoretical perspective, it is crucial to evaluate and analyze the ecological balance of cultivated land from a spatial-temporal standpoint. This introduces a dynamic equilibrium model that recognizes the nexus between ecological compensation and cultivated land change as dynamic, interconnected systems. Moving forward, the novelty of this study may combine socio-ecological and climatic resilience theories to analyze the interplay between social and ecological systems in the context of cultivated land protection and ecological compensation. These theoretical innovations can contribute to a more nuanced and adaptive understanding of the prerequisite relationship between ecological balance and cultivated land, guiding practical approaches for sustainable land use and conservation. Therefore, this paper chooses Northeast China as its research region and defines the scope of the CLEC and the ecological service function that can be used to clarify compensation standards. In addition, this investigation considers the influence of socioeconomic factors, such as cultivated land area, population, GDP, etc., on the compensation standard. Furthermore, this paper adopted an accounting model to make the compensation standard more realistic.

This paper used a case study in NC to carry out a spatio-temporal analysis of the CLEC from 2000 to 2020. First, we used the ecological footprint model to calculate the regional ecological footprint (EF) of the cultivated land and its ecological carrying capacity (EC). Second, we calculated the ESV based on a modified equivalence factor. Then, we built a CLEC model based on the ecological footprint model and the ESV and measured the amount of CLEC required for each city in NC. Finally, we put forward CLEC's management policy recommendations. This study reveals CLEC's spatial-temporal characteristics and provides policy recommendations for sustainable cultivated land protection and utilization.

## 2. Materials and Methods

### 2.1. Study Area

Northeast China is one of the four black soil areas in the world, including 41 cities in Heilongjiang, Jilin, Liaoning, and the five eastern league cities of Inner Mongolia. NC is located at latitudes between 38°43′ and 53°33′ N and longitudes between 118°53′ and 135°05′ E (Figure 1). It strategically occupies a central position between Liaohe Plain and Songnen Plain, surrounded by the Greater Khingan, Lesser Khingan, and Changbai

Mountains on three sides. The Songhua River flows between the Lesser Khingan and Changbai Mountains, leading to the Three Rivers Plain. Moreover, NC experiences a temperate continental monsoon climate. Most of this precipitation occurs during the growing season, from April to September. As of 2020, the cultivated land in NC covered $35.87 \times 10^6$ ha, accounting for 26.6% of the country's cultivated land area, and the overall value of its agricultural output reached CNY 673.25 billion, accounting for one quarter of the country's total agricultural output. NC has a natural geographic environment suitable for the growth of various crops such as soybeans, corn, and rice, laying a good foundation in terms of resources and the environment for the development of NC. However, large amounts of fertilizers and mechanization have been increased to produce more grain [41]. Sustained intensive application has reduced the quantity and quality of the cultivated land and caused the ecosystem of the cultivated land to deteriorate [42]. As such, this study chooses NC to analyze CLEC from a spatio-temporal perspective. This research is likely to be effective in enabling the government to restore the balance between agricultural production and ecological preservation.

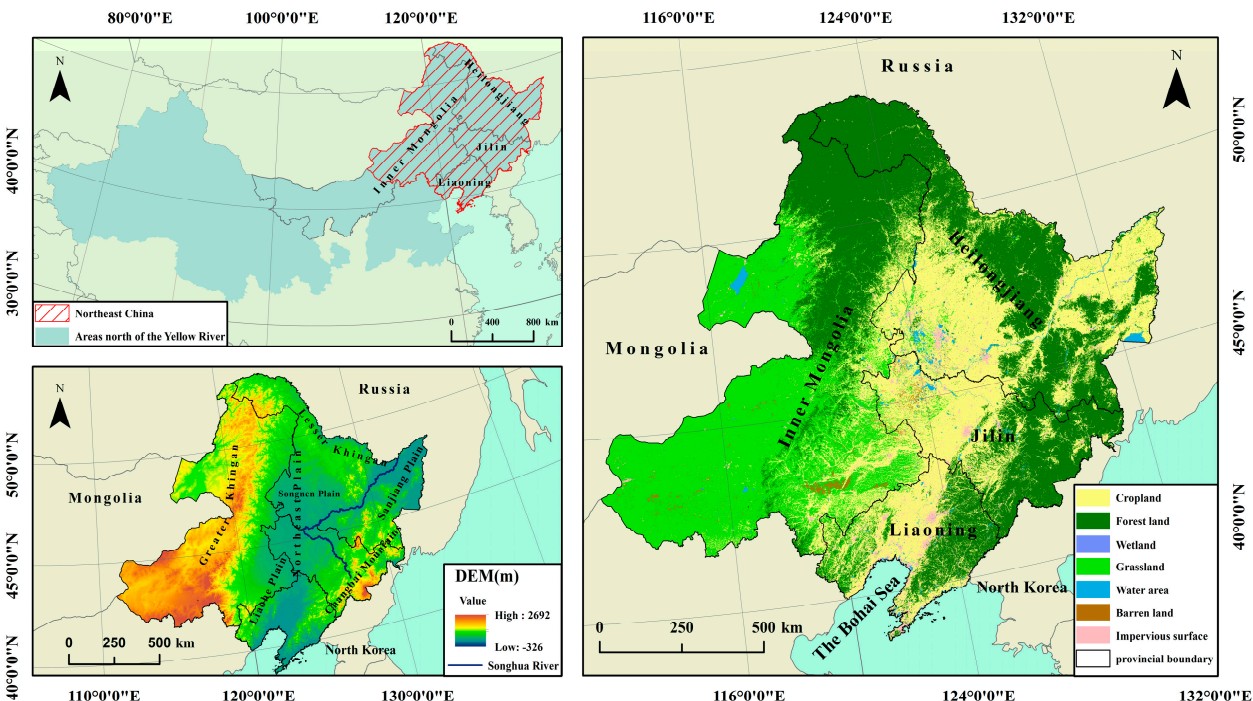

**Figure 1.** Map of Northeast China.

## 2.2. Data Sources

The data included land and economic data. The land use data for NC were obtained from the Data Center for Resources and Environmental Sciences, Chinese Academy of Sciences (RESDC) (http://www.resdc.cn, accessed on 18 August 2023). The data were produced from Landsat TM/SEM remote sensing imagery. As such, the spatial resolution was $30 \times 30$ m, and the comprehensive accuracy was higher than 93% [43,44]. The data on cultivated land production, the consumption of the populations of the 41 cities, and other economic data for NC were taken from the statistical yearbooks of each city. Specifically, they include the sown area of crops; total production; production; size; consumption; average price of rice, soybean, and corn grain crops; the total population; and the level of urbanization and development. Table 1 shows all the symbols used in the equations.

**Table 1.** Symbols, explanations of the data, and units.

| Symbols | Explanation | Unit |
|---|---|---|
| $i$ | Rice, maize, soybean | / |
| $o_i$ | The regional per capita production of food crop $i$ | kg/person |
| $m_i$ | The sown area of food crop $i$ | ha |
| $p_i$ | The mean cost of food crop $i$ across the nation | CNY/kg |
| $q_i$ | The output or production of food crop $i$ | kg/ha |
| $M$ | The area of cultivated land for food crop $i$ | ha |
| $En$ | Engel coefficient for urban and rural areas | % |
| $p_i$ | The national average yield for crop $i$ | kg/ha |
| $a$ | Biologically productive land area of cultivated land per capita | ha/person |
| $r$ | Equivalence factor | / |
| $y$ | Yield factor | / |
| $N$ | The total population of the region | Number |
| $GDP$ | GDP of China | CNY |

### 2.3. Research Mechanism

In the process of the market economy's operation, because the private marginal costs and benefits and social marginal costs and benefits are not equal, relying on competition makes it difficult to achieve the maximization of the welfare of the entire social space when externalities appear; the government sector must compensate for the difference between the two to a certain extent, in order to realize the internalization of its external effects. As the essential material carrier of agricultural production, cultivated land is vital in sustaining economic and social development. It provides various ecological services such as regulation, supply, support, and cultural services. Analyzing this from the perspective of the supply and demand of these ecological services (Figure 2), considering only the marginal private benefits (demand for ecological services of cultivated land) for farmers in carrying out ecological protection of cultivated land, the equilibrium point between it and the cost of protection (supply of ecological services of cultivated land) is C; and taking into consideration the ecological and social benefits of arable land resources at the same time, the equilibrium point is D. And the ecological services of cultivated land at point C and the demand for the ecological services of cultivated land are smaller than that of point D. In the case of the externality of cultivated land's public goods, according to the principle of utility maximization that marginal cost equals marginal benefit, the actual supply of ecological services will be less than the social demand for them, and in order to increase the supply of services, it is necessary to compensate for the ecological externality of cultivated land, so as to make the curve of the marginal private benefit shift upward and approach the marginal social benefit, and to reach the equilibrium of supply and demand at point D.

In terms of monetizing the ecological services provided by cultivated land, area A is the value of the ecological services consumed by farmers themselves for cultivated land protection, B is the value of the ecological services consumed by society as a whole, and B-A is the value of residual ecological services provided by farmers to the society after removing their consumption. In order to incentivize the protection behavior of the ecological service suppliers of arable land, they should be compensated for the portion of the loss they suffered. However, ecological services such as raw material supply, food production, can realize their economic value through market transactions. They should not be counted as the content of the compensation. Therefore, in this research, the equivalent factor method was applied to measure the total ecosystem service value and non-market value of the study area, combined with an ecological footprint model to measure the self-consumption coefficients of the ecological service suppliers of the cultivated land and the consumption of non-market value, and to finally obtain the residual portion of the non-market value of the cultivated land ecosystem services as the CLEC.

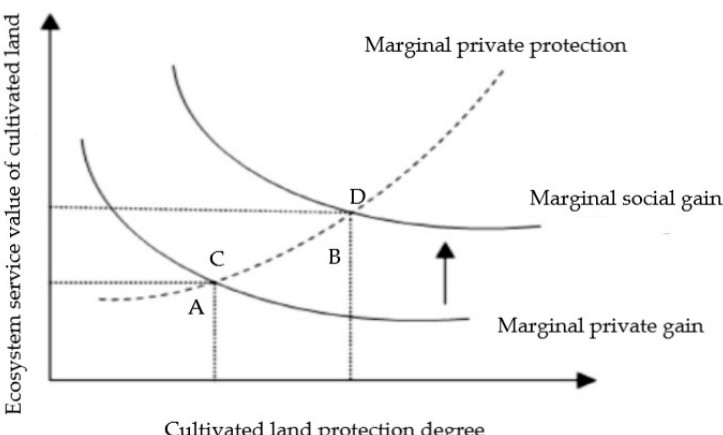

**Figure 2.** Economic analysis of ecological compensation for cultivated land. Notes: A and B are the areas of the rectangles formed by the perpendiculars to the horizontal and vertical axes; C and D are the intersections of the marginal private cost of conservation with the marginal social benefit and the marginal private benefit, respectively.

### 2.4. Methods

Effective implementation of CLEC requires accurately measuring compensation standards [9,17]. To accurately measure CLEC, we combined the ecological footprint model and the value of the ecosystem services to measure the CLEC in NC (Figure 3). As shown in Figure 2, our measurement of CLEC included three parts. Firstly, this study measured cultivated land's ecological footprint and carrying capacity to assess the human impacts on cultivated land ecosystems and the environment in NC. Secondly, this study calculated the ecological surplus and the ecological deficit of cultivated land. If the cultivated land's ecological footprint outweighed its ecological carrying capacity, the ecological surplus can provide an ESV to compensate for the ecological deficit. Conversely, an ecological deficit provides monetary compensation for the ecological surplus. Finally, the research calculated the amount of CLEC due to different cities in NC.

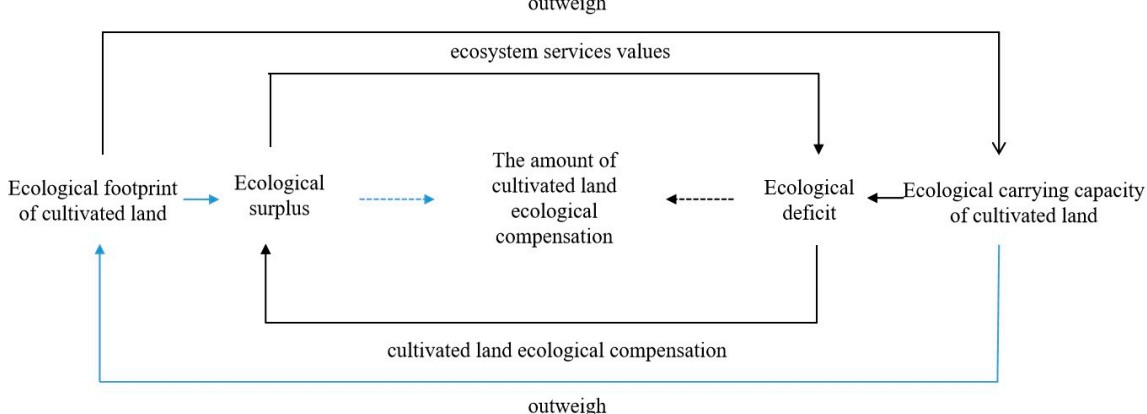

**Figure 3.** Measurement of CLEC.

### 2.4.1. The Ecological Footprint of Cultivated Land

The ecological footprint quantifies the human appropriation of environmental resources. The ecological footprint of cultivated land accounts for the human utilization of cultivated land resources, balancing the differences in the production capacity of different land types through balancing the factors and converting various types of land into the same ecological production area, reflecting the location of the cultivated land required for human

survival, consumption, and abandonment in the region. The equation for calculating this is outlined below [31]:

$$EFs = N \times ef = N \times \sum_{i=1}^{n} rA_i = N \times \sum_{i=1}^{n} r\frac{O_i}{P_i} \qquad (1)$$

where *EF* is the ecological footprint of cultivated land; *N* is the total regional population (persons); *ef* denotes the per capita EFs; and *r* is the equivalence factor of cultivated land, which reflects the potential productivity of a particular type of ecologically productive cultivated land, a natural attribute of cultivated land. This study used the Chinese cultivated land equivalence factor of 1.74 as the equivalence factor for the EFs, calculated by Liu and Li [45]. *i* is a specific type of crop, $A_i$ is the individual share of ecologically viable cultivated land area for crop *i*, $O_i$ is the regional per capita production of crop *i* (kg/person), and $P_i$ is the national average yield for crop *i* (kg/ha).

### 2.4.2. The Ecological Carrying Capacity of Cultivated Land

The ecological carrying capacity includes the self-sustaining and regulating part of the ecosystem and the part that absorbs and supports the pressure of human economic activities. The ecological carrying capacity of cultivated land refers to the maximum biologically productive area of cultivated land that a region can offer while maintaining a good ecosystem. The ecological carrying capacity of cultivated land refers to the capacity of cultivated land to sustain human activities [46]. The equation for calculating this is outlined below:

$$ECs = N \times a \times r \times y \times (1 - 12\%) \qquad (2)$$

where *EC* is the ecological carrying capacity of cultivated land, *a* is the biologically productive area of cultivated land per capita (ha/person), and *y* is the cultivated land yield factor, which is the ratio of the capacity of a given ecosystem type to provide a given ecosystem service in the region to the national average for that ecosystem service. In calculating the EC, this study adopted the yield factor of agricultural land for each province of China calculated by Liu et al. as the yield factor of EC (Table 2) [47]. In addition, the report Our Common Future by the World Commission on Environment and Development suggested allocating 12% of productive land area for the preservation of biodiversity in calculations of ECs [48].

**Table 2.** Yield factors in Northeast China, calculated by Liu et al. [45].

| Province | Inner Mongolia | Liaoning | Jilin | Heilongjiang |
|---|---|---|---|---|
| Yield factor | 0.52 | 0.84 | 0.83 | 0.58 |

### 2.4.3. Measurement of the ESVs of Cultivated Land

Constanza [49] initially proposed the equivalent factor of ecosystem services and evaluated the global ESVs. Subsequently, drawing on China's specific circumstances, Xie et al. [50] improved the equivalent value of the ecosystem services per unit of cultivated land area (Table 2). The ESVs of cultivated land refers to the ecological impact of utilizing cultivated land on the surrounding environment, including water conservation, climate regulation, and the maintenance of biodiversity. The equivalent factor of the ESVs of cultivated land is the economic value of the food production service per unit of area of cultivated land per year, using 1/7 of the market value of the average crop yield as the monetary value of each unit of the equivalent factor of ESVs. The formulae are as follows:

$$ESVs = ESV_i \times S_i \qquad (3)$$

$$ESV_i = E_a \times F \times Z_i \qquad (4)$$

$$E_a = \frac{1}{7} \sum_{i=1}^{n} \frac{m_i \times p_i \times q_i}{M} \tag{5}$$

$$Z_i = X_i \times Y_i = \frac{B_i}{C_i} \times \frac{1}{1 + e^{-t}} \tag{6}$$

$$t = \frac{1}{E_n} - 3 \tag{7}$$

where $ESV$ shows the total ESVs of cultivated land (CNY), $ESV_i$ is the ESVs of the cultivated land per unit of area (CNY), $E_a$ is the ESV of the unit's equivalent factor (CNY/ha), $Z_i$ is the economic adjustment factor for cultivated land ecosystems, $F$ is the corrected equivalent factor (Table 3), $S_i$ is the total area sown with grain in an area (ha), $n$ is the total number of food crops, $i$ is the type of food crop (the main food crops in NC are rice, maize, and soybean), $m_i$ is the sown area of crop $i$ (ha), $p_i$ is the national average price of crop $i$ (CNY/kg), $q_i$ is the yield of crop $i$ (kg/ha), $M$ is the total sown area of the $n$ types of food crops (ha), $En$ is the combined urban and rural Engel coefficient for the region, 1/7 signifies the economic worth generated by natural ecosystems in their pristine state without human interference [51], $B_i$ is the local per capita GDP (CNY), and $C_i$ represents the national per capita GDP (CNY).

**Table 3.** The ESV per unit of area of cultivated land in Northeast China.

| Primary Types | Secondary Types | Equivalent Factor |
|---|---|---|
| Provision | Food manufacturing | 1.00 |
| | Raw material production | 0.39 |
| Regulation | Gas regulation | 0.72 |
| | Climate regulation | 0.97 |
| | Water conservation | 0.77 |
| | Waste disposal | 1.39 |
| Support | Soil formation and retention | 1.47 |
| | Biodiversity | 1.02 |
| Culture | Aesthetic landscape | 0.17 |

2.4.4. Calculating the Ecological Compensation for Cultivated Land

Calculating the standard of CLEC requires considering several factors. These include disparities in the region's ecological environment, the residents' consumption of ecological resources, and their willingness to contribute towards ecological compensation. Consequently, this study integrated the ecological footprint model and the ESVs to calculate CLEC. These were coupled with adjustments based on regional socioeconomic development. Moreover, recognizing variations in the actual capacity for compensation among regions and adjusting the region's actual compensation capacity was essential, as the actual compensatory capacity varied from region to region [31]. In line with this, this study established a model of the ecological compensation for cultivated land.

$$V = ET \times ESV_i \times R_i \tag{8}$$

$$ET = \frac{EF - EC}{EC} \tag{9}$$

$$Ri = \frac{e^{En} \times GDP_i}{(e^{En} + 1) \times GDP} \tag{10}$$

where $V$ is the total amount of CLEC (CNY/year), $R_i$ is the coefficient of CLEC in the region, $ET$ is the ecological overload index, $GDP_i$ is the total output in the current year (CNY), and $GDP$ is the aggregate output of NC (CNY).

## 3. Results

### 3.1. The Ecological Footprint of Cultivated Land in Northeast China

In terms of temporal variations, the EF presented a fluctuating growth trend in NC from 2000 to 2020. In 2000, 2005, 2010, 2015, and 2020 (Figure 4). The total EF of NC was $235.72 \times 10^5$ ha, $345.77 \times 10^5$ ha, $582.42 \times 10^5$ ha, $449.83 \times 10^5$ ha, and $524.55 \times 10^5$ ha, respectively. Over 20 years, the total EF in NC increased by $288.83 \times 10^5$ ha, and the rate of increase was 55.06%. While there was a decrease in the total EF in 2015, there was a general upward trend, which was particularly noticeable between 2005 and 2010. According to the total EF, there was a relatively small fluctuation from 2015 to 2020, and the magnitude of the variation was 14.24%. However, there was a marked increase in the EF between 2005 and 2010, mainly caused by increased food production.

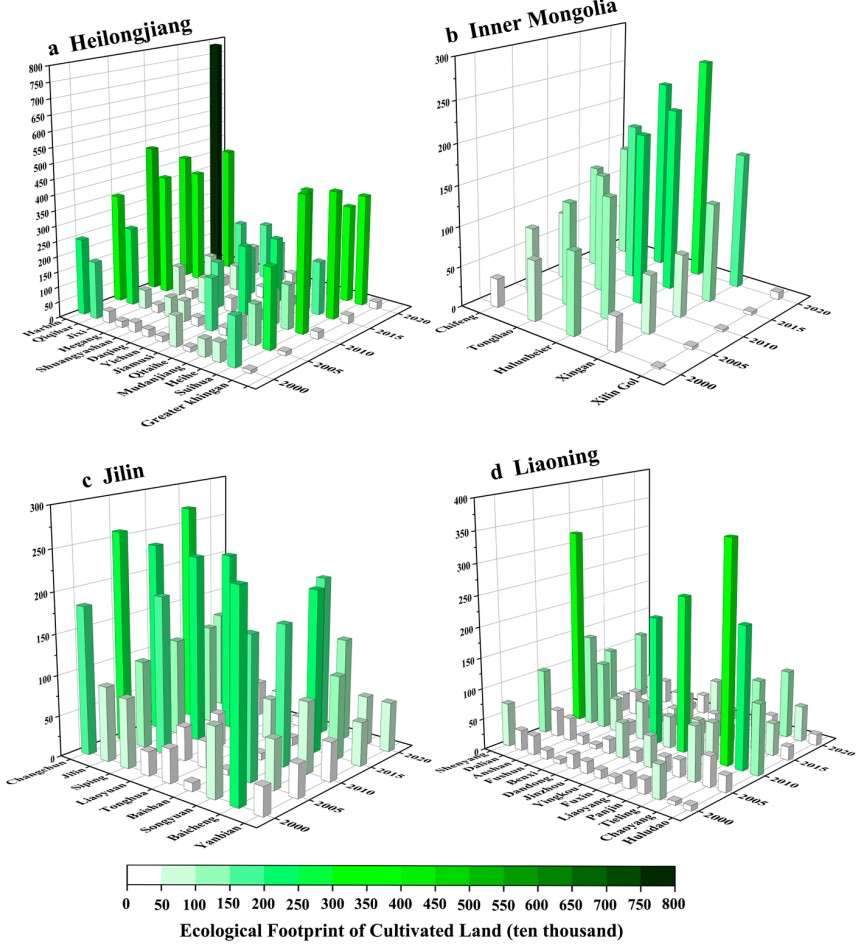

**Figure 4.** Temporal distribution of the EFs in Northeast China.

The spatial distribution of EFs was also critical. With GIS mapping software, we could visually observe and analyze the spatial distribution of EFs in NC (Figure 5). In terms of spatial distribution, the high-value areas of the EF exhibited a broad distribution and reached a maximum in 2010. The areas with a larger EF were in the northern parts of NC. The western part of this region predominantly contained areas with a lower EF. The EFs in the five eastern leagues of Inner Mongolia showed an increasing trend. However, that of the Xilingol League was the lowest in the whole of NC, with a value of $0.93 \times 10^5$ ha in 2020. In Liaoning Province, the EF in Tieling City grew significantly high and quickly. From 2000 to 2020, the indicator surged from $5.39 \times 10^5$ ha to $11.06 \times 10^5$ ha, with a growth rate of 51.27%. The EF of Jilin Province has risen, but the growth trend has been relatively stable over the past two decades. In this province, the EF of Baishan has significantly increased from $1.03 \times 10^5$ ha in 2000 to $19.78 \times 10^5$ ha in 2020. Among all prefecture-level cities in

Heilongjiang Province, EFs have generally been on the rise and have continued to maintain a high value.

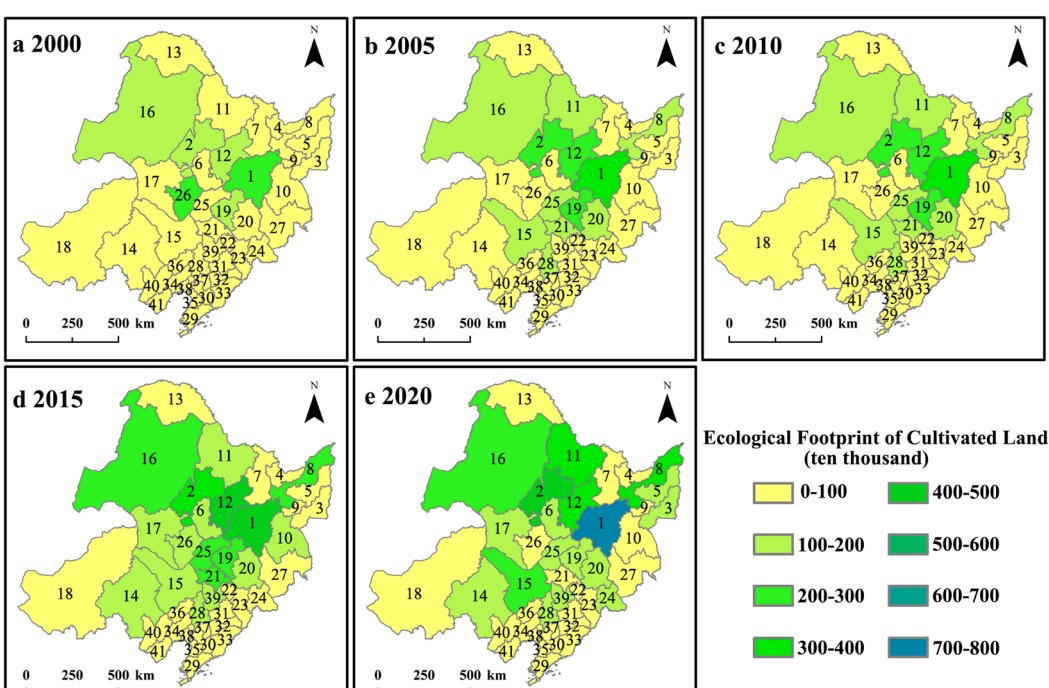

**Figure 5.** Spatial distribution of the EFs in Northeast China.

*3.2. Ecological Carrying Capacity of Cultivated Land in Northeast China*

In terms of temporal variation, the EC demonstrated a fluctuating growth trend in NC from 2000 to 2020. From 2000 to 2020 (Figure 6). the total EC in NC increased by $131.37 \times 10^5$ ha, and the rate of increase was 37.47%. However, the EC had a significant drop in 2015, decreasing by $103.78 \times 10^5$ ha (compared with 2000). Although the total EC declined in 2015, it demonstrated an upward trend overall; in particular, from 2005 to 2010 the rate of increase was 63.54%. In the long run, a highly concentrated production pattern disrupted the moderate equilibrium of the cultivated land's ecosystems.

In the spatial distribution map of the EC from 2000 to 2020 (Figure 7) as a whole, the EC showed growth from southwest to northeast. In NC, the capital cities of the provinces, such as Harbin, Changchun, and Shenyang, had the highest ECs within the region. However, Greater Khingan and the Xilingol League showed a lower EC, especially Greater Khingan. In 2020, the EC was only $1.73 \times 10^5$ ha in Greater Khingan. Moreover, the ecological status of cultivated land varied widely in Liaoning Province. According to the EC, Liaoning Province can be categorized into three zones: a high-value region including Chaoyang City, Fuxin City, Huludao City, Jinzhou City, Shenyang City, and Tieling City; a central low-value region encompassing Dalian City, Anshan City, and Dandong City; and "ecological consumption areas" and "ecological service areas," which dominate the western part of Liaoxi, the northern section of Liaocheng, and the southern reaches of Liaonan.

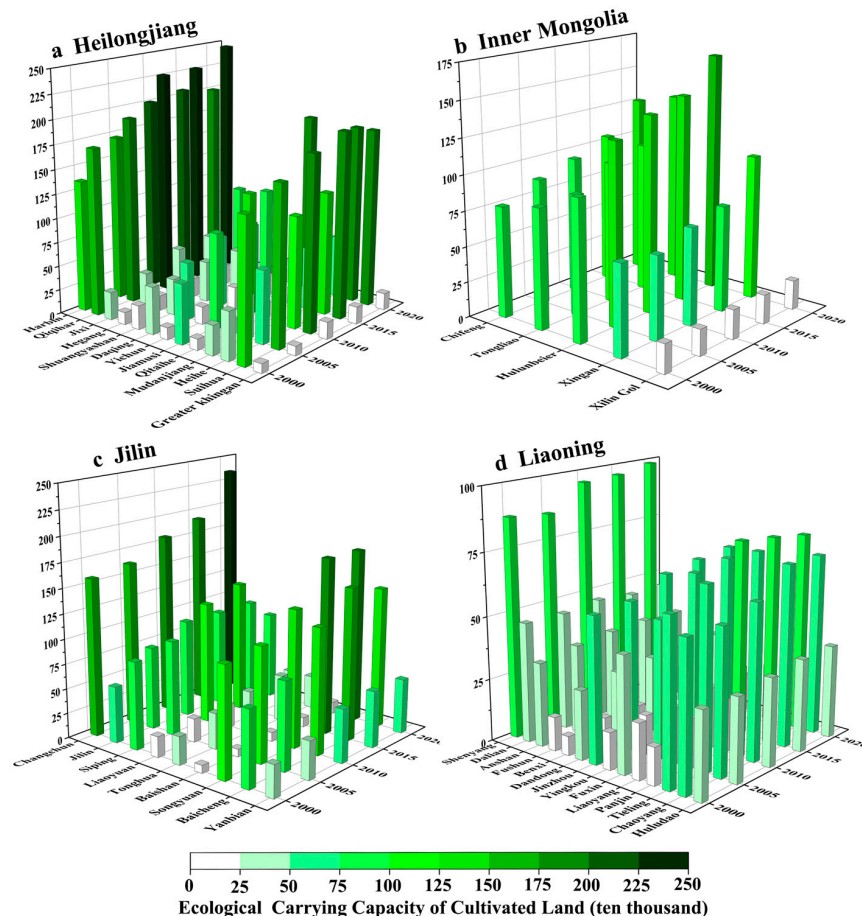

**Figure 6.** Temporal distribution of the ECs in Northeast China.

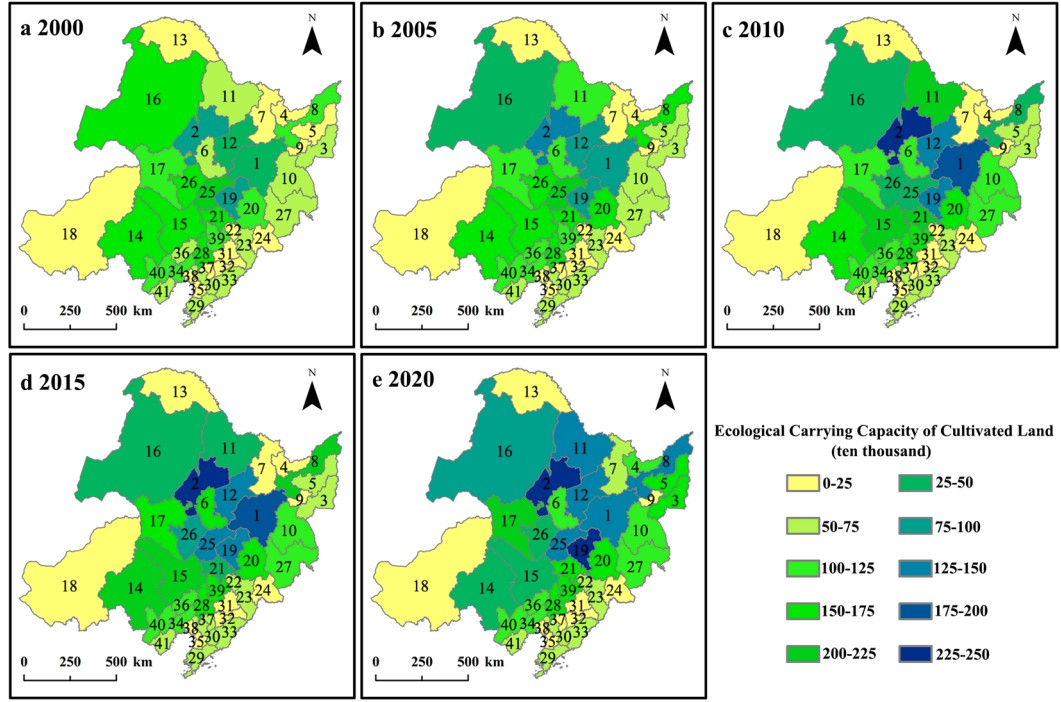

**Figure 7.** Spatial distribution of the ECs in Northeast China.

### 3.3. Value of the Ecosystem Services of Cultivated Land in Northeast China

Between 2000 and 2020, the ESV in NC exhibited a pattern of initial growth, followed by a decline and then a subsequent rise. In 2000, 2005, 2010, 2015, 2020, the total ESV of NC was CNY 109.39 billion, CNY 130.35 billion, CNY 222.65 billion, CNY 111.58 billion, and CNY 123.03 billion, respectively. Over 20 years, the total ESV in NC increased by CNY 13.64 billion, a year-on-year increase of 0.6% (compared with 2000). Despite a decrease in the total ESV in 2015, an overall upward trend was observed. Notably, there was a substantial surge in the ESV between 2005 and 2010, representing an increase of over 42%. However, the ESV was significantly lower between 2010 and 2015. This is because in 2015, NC produced less food, which resulted in a decrease in its ESV.

According to the spatial distribution map of the ESV from 2000 to 2020 (Figure 8), it can be seen that the ESV of cultivated land showed growth from southwest to northeast. Simultaneously, the spatial distribution of the ESV exhibited an uneven pattern of development. Some regions, such as Hinggan League, Baishan City, Jixi City, and Heihe City, presented a rising trend in their ESVs. In contrast, Shenyang City, Baicheng City, and Qiqihar City presented a declining trend. The execution of an ecological protection policy did not lead to significant improvements in mitigating the uneven distribution of the ESV within NC. The impact on diminishing regional disparities was marginal. Among the 41 cities in NC, the ESVs of the cultivated land in the five eastern leagues of Inner Mongolia, Da Hinggan Ling Prefecture, Yanbian Korean Autonomous Prefecture, and Yichun City was relatively small, with the Xilingol League having the lowest value (CNY 0.03 billion). In addition, Changchun City, Shenyang City, and Harbin City had the highest ESVs. As provincial capitals, these three cities have a high level of socioeconomic development, and their residents have greater willingness and ability to contribute to ecological compensation, compared to the other cities in NC. Heilongjiang Province, Mudanjiang City, and Da Hinggan Ling Prefecture showed a decreasing trend in the ESVs of their cultivated land, while all other cities presented a rising trend from 2000 to 2020. In Jilin Province, except for Changchun City, the ESV presented a decreasing trend. In particular, Baicheng City declined from a high-value area in 2000 to a low-value area in 2020. As a major agricultural province in China, Heilongjiang Province guarantees national food security and its population has a rising awareness of cultivated land protection.

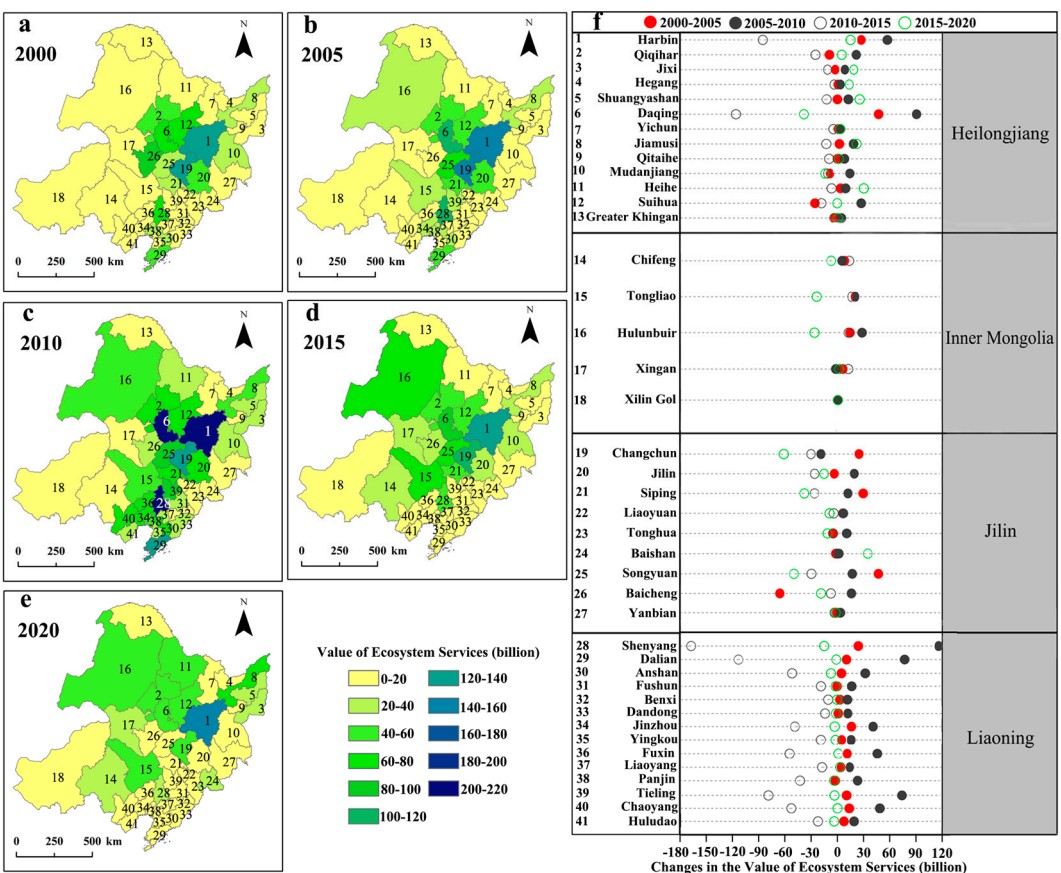

**Figure 8.** Spatial and temporal distribution of the ESVs in Northeast China.

### 3.4. Amount of Ecological Compensation Needed for Cultivated Land in Northeast China

The CLEC in the 41 cities of NC was calculated using Equation (8). In terms of the amount of CLEC, NC exhibited a pattern of initial growth, followed by a decline, and then a subsequent resurgence from 2000 to 2020. Over 20 years, the total CLEC in NC increased by CNY 1.8 billion and the rate of increase was 12% (compared with 2000). From 2000 to 2010, the CLEC significantly increased from CNY 14.77 billion to CNY 23.24 billion. However, after 2010, CLEC significantly decreased in various municipalities. In 2015, the CLEC was only half as much as it had been in 2010 in Jilin City, Harbin City, and Suihua City. As such, if action towards CLEC is not accelerated, the amount of compensation will subsequently decrease. Then, the motivation of farmers to produce food and protect cultivated land will also be reduced further. This situation will indirectly affect the sustainability of cultivated land's ecology and food security.

Figure 9 illustrates the spatial distribution of CLEC from 2000 to 2020. The spatial distribution showed an increasing trend from southwest to northeast. Moreover, the total amount of CLEC varied significantly within the region. Across the whole study area, except for Chifeng City, Tongliao City, Hulunbeier City, and Hinggan League, all locations were ecological compensation areas. Heilongjiang Province, Yichun City, and Daxinganling City showed a decreasing trend and had lower amounts of ecological compensation: only CNY 11.58 thousand and CNY 42.62 thousand, respectively. This shows that the CLEC of NC was much higher than the national average for cultivated land.

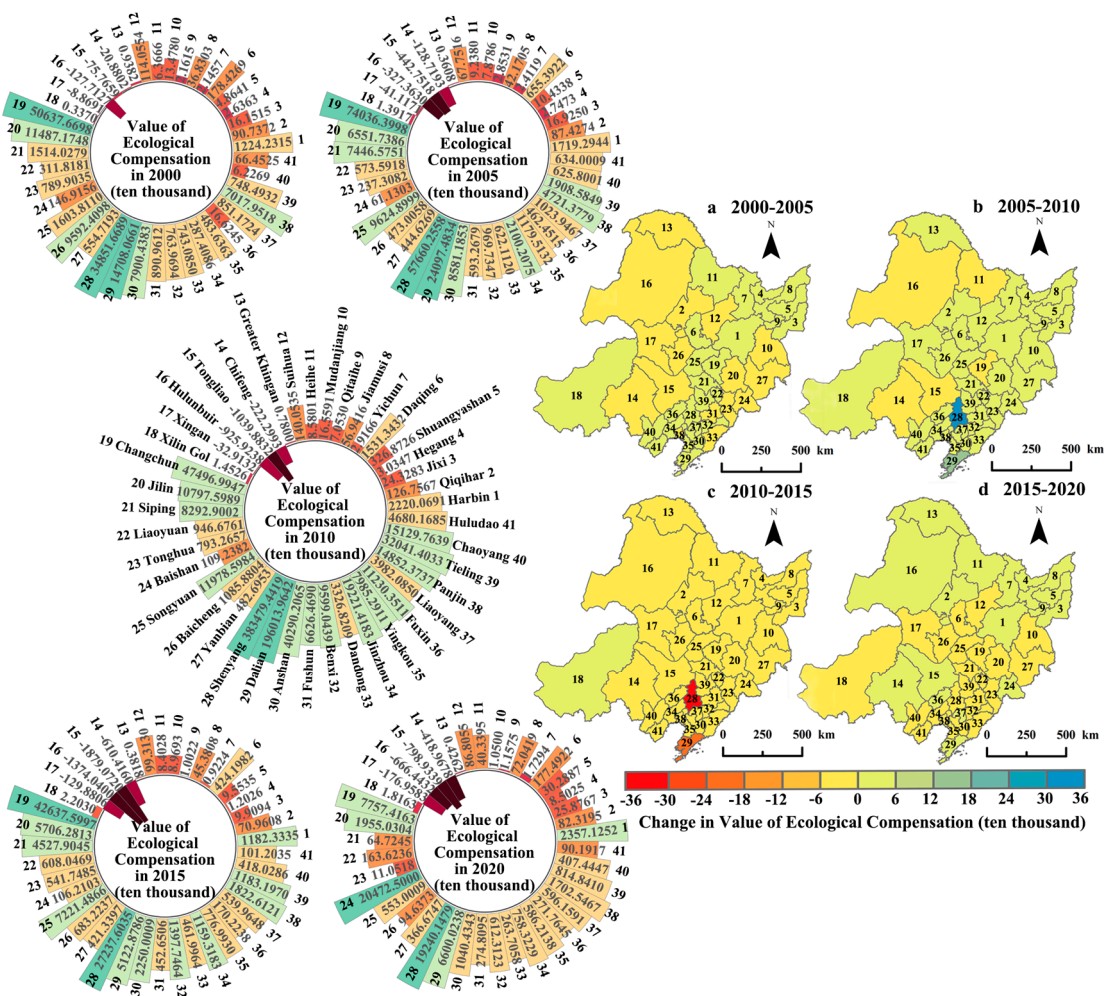

**Figure 9.** Spatial and temporal distribution of CLEC in Northeast China.

## 4. Discussion

### 4.1. The Rationality of Ecological Compensation for Cultivated Land

The research measured CLEC based on EFs and ECs, unlike some approaches that used only ESVs [21,52]. Determining standards for ecological compensation is the core of CLEC, because it has significant implications for the long-term use of cultivated land ecosystems [24,53]. The results obtained from the ecological footprint model revealed a positive correlation between the distribution of the ecological footprint, the population size, and the economic scale. This observation aligns with previous research [32,54]. Currently, NC heavily relies on consuming its natural capital stock, leading to unsustainable development. Consequently, implementing ecological compensation mechanisms is crucial for mitigating the regional imbalances in both the ecological environment and the economy.

Between 2005 and 2010, there was a noteworthy surge in the ESV within NC, which increased by over 42%. This rise can be attributed to the government's initiative in establishing an ecological protection framework for cultivated land, aiming to achieve balanced development between the economy and the environment. This has made it possible to increase the ESV in the region by adopting environmental restoration measures for cultivated land. The market component of the ESV undergoes direct conversion into currency within market mechanisms, thereby fostering the region's economic and social development. The northeast region of NC had a high ESV; however, the western regions had a lower ESV. The areas with the highest value in terms of CLEC are primarily clustered in the central agricultural production zones of NC. These areas are crucial for extracting resources and

intensive agricultural production in NC, making them prime examples of zones of high ecological consumption.

Previously, some scholars affirmed that socioeconomic and natural factors have the most significant explanatory influence on the ESV of cultivated land [55,56]. The resources of cultivated land should be strengthened and protected in the future, and the EC should avoid the emergence of a vicious cycle. With the development of the social economy, the residents' demand for land and their ability to pay will increase the occupation of cultivated land, triggering more severe conflicts between people and the land. To a certain extent, this also contributes to the ESV of each piece of cultivated land in the compensation zone, thus increasing the total amount of compensation.

Ultimately, this study has developed a city-level ecological compensation model for NC. The results of this study are basically in line with the reality of NC. This offers a new approach to implementing compensation and serves as a valuable reference for enhancing CLEC policies.

*4.2. Recommendations for the Ecological Compensation of Cultivated Land in Northeast China*

4.2.1. Transforming Government-Oriented Approaches and Introducing Market Mechanisms

The current government-oriented mechanism for CLEC is constructed so that the government plays a leading role in the CLEC, and this method applies to China's current national conditions. However, to ensure the sustainability of CLEC, it is necessary to actively transition to an ecological compensation system with market allocation at the heart so that the resource market can become a bridge between the ecological value and ecological compensation, and gradually establish a market-oriented and diversified mechanism of ecological protection. Meanwhile, the government has also encouraged the public to participate in ecological protection and the development of black soil. The PPP (public–private partnership) approach was implemented to mobilize reimbursement funds. It considered the implementation of policies such as environmental pollution liability insurance, explored modes of financing such as balancing sources of income and ecological expectations, and encouraged eligible enterprises to participate in long-term investments. Implementing third-party governance would attract social capital to engage in CLEC in NC. On the whole, it is necessary to establish a government-led mechanism of compensation for ecological protection, with the organised participation of society and effective regulation by the market.

4.2.2. Diversify the Approach to Ecological Compensation for Cultivated Land

A singular CLEC approach typically fails to fully satisfy the requirements for the ecological protection of cultivated land and the provision of ESV. This can result in the neglect of crucial aspects of ecological protection. As such, exploring various modes of ecological compensation during implementation is essential. Given the scale dependencies of stakeholders in different regions, it is vital to enhance collaborative governance and platforms for transactions related to the compensation of ecological services for protecting grain farmland. At the prefecture and municipal level, the prefecture pays the compensation funds into the specific account of the arable land management platform. Through the coordination and integration of the management platform, the amount of compensation for the compensated municipalities is determined, and the corresponding funds are allocated, realizing horizontal and vertical financial transfers between the ecological compensation of arable land in the municipalities. Establishing a mutually beneficial cooperation mechanism is crucial to ensuring the sustainability of both national food security and the utilization of farmland. More importantly, NC should explore "blood-making" compensation to improve the ecological capacity of local cultivated land. Therefore, in light of the funding for compensation, we should explore comprehensive compensation methods, such as those based on policy, technical knowledge, and talent.

### 4.2.3. Determine Reasonable Standards of Compensation for Cultivated Land and Promote Positive Interactions among Stakeholders

Assessing loss is an essential basis for determining the standards of compensation. On the one hand, we should determine the level of the ecological function of cultivated land through soil testing, land grading, satellite and radar positioning, etc. We should then set up a database of information about cultivated land to assess the loss in the ecological value of the cultivated land. The results of such assessments should be combined with revising standards for compensation. On the other hand, we can incorporate assessments of performance in protecting cultivated land and ecological compensation into the government's performance assessment system and regularly assess the changes in locally cultivated land during the tenure of relevant leaders and government officials. Meanwhile, to promote the rule of law as a system for protecting cultivated land and ecological compensation, it is necessary to clarify the target of compensation, the compensation period, the standard for compensation, the method of compensation, and the rules for its implementation. We also need to set up the corresponding supervisory institutions and clarify the rights and obligations of the relevant interested parties. In addition, CLEC follows the principle of "who benefits, who compensates, who pollutes, who pays," in which the corresponding ecological compensation for protecting cultivated land is paid to those whose interests are damaged, maximizing the fairness of the distribution of the interests of the main body of the cropland expropriated for protection, as well as accounting for the externalities of the cultivated land ecosystem, promoting benign interactions among the stakeholders and providing economic incentives for the protectors of the ecosystem.

### 4.3. Limitations and Future Prospects

This study focused on the government's compensation mechanism for the regional ecological protection of cultivated land on the city scale, i.e., an area-wide problem of ecological compensation. Ecological protection is not only the responsibility of city-scale regional administrative agents, but also includes individuals, communities, and other subjects of the actions taken to protect the ecological environment of cultivated land. Microscopic habits for protecting the ecology of functioning cultivated land and protection projects still need to be explored further, as well as the market mechanism of CLEC. This study considered the area of cultivated land, the population, and GDP, and utilized the yield data of crops in the municipalities of the study area instead of the data on people's consumption of biological resources. It did not consider factors such as geographical location and socioeconomic development, which will make the results of the study biased, so improving the methodology used to increase the accuracy of the CLEC is still a direction for future research.

### 5. Conclusions

Taking Northeast China as the research region, this study analyzed the spatio-temporal changes in CLEC from 2000 to 2020. This study introduced an ecological footprint model combined with the ESV to study CLEC, which allowed us to determine the standards for compensation related to CLEC and enriched the existing research. The main conclusions are as follows: The EF and EC grew from southeast to northeast, increasing by $288.83 \times 10^5$ ha and $131.37 \times 10^5$ ha, respectively. There was a substantial surge in the ESV between 2005 and 2010, and the ESV reached its highest point in 2010 (CNY 222.65 billion). From 2000 to 2020, the total CLEC required for NC was CNY 14.77 billion, CNY 20.63 billion, CNY 23.24 billion, CNY 10.34 billion, and CNY 16.55 billion. The influence of population and area on the standards for compensation in different places were quite different.

Consequently, this study can provide some suggestions for policymakers implementing CLEC. The Chinese government promotes the healthy development of China's grain market, provides certain subsidies to farmers, and encourages them to continue to engage in agricultural work. Therefore, this study has particular significance and value for CLEC in practice. However, there are some limitations. Our investigation only con-

sidered three factors (area of cultivated land, population, and GDP) in the standards for compensation. In the future, we should consider the factors of geographical location and socioeconomic development. Moreover, this study did not consider the self-consumption and spillover effects of a region on the ESV when determining the standards for compensation. Because different areas will consume part of their ecosystem services in the process of development, there are spillover effects in their ecosystem services, so determining how to include self-consumption and the spillover effects of ecosystem services in the standards for compensation will also be an exciting topic for future research.

**Author Contributions:** Conceptualization, L.W. and Q.L; methodology, L.W.; validation, L.W., B.F. and Q.L.; formal analysis, L.W.; investigation L.W., Y.L. and B.F.; resources, Q.L.; data curation, Y.L.; writing—original draft preparation, L.W.; writing—review and editing, Q.L.; visualization, B.F.; supervision, Q.L.; project administration, Q.L.; funding acquisition, Q.L. All authors have read and agreed to the published version of the manuscript.

**Funding:** This research was funded by the China Postdoctoral Science Foundation, grant number 2021M700738, the China Postdoctoral Science Foundation, grant number 2022T150103, and the Northeast Agricultural University Academic Backbone Program, 21XG52.

**Institutional Review Board Statement:** Not applicable.

**Informed Consent Statement:** Informed consent was obtained from all subjects involved in the study.

**Data Availability Statement:** The data presented in this study are available upon request from the corresponding author. The data are not publicly available due to privacy or other restrictions.

**Conflicts of Interest:** The authors declare no conflict of interest.

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
