# Peer review of "A Spatio-Temporal Analysis of the Ecological Compensation for Cultivated Land in Northeast China"

_land, doi:10.3390/land12122179_

Round 1
Reviewer 1 Report
Comments and Suggestions for Authors
I did not see the author's innovation in this manuscript. The author only addresses the issue of compensation for arable land in a region. The author need to make certain improvements to the ecological compensation method in combination with specific practices, rather than the application of traditional methods.
Comments on the Quality of English LanguageModerate editing of English language required.
Reviewer 2 Report
Comments and Suggestions for Authors
Manuscript Title: Spatial-temporal Analysis of Cultivated Land Ecological Compensation in Northeast China
Manuscript ID: land-2669247
General Comments:
The manuscript's intent is to conducts a spatial-temporal analysis of cultivated land ecological compensation in Northeast China with considering both ecological footprint and ecosystem service value. This paper used the ecological footprint model to calculate the ecological footprint and ecological carrying capacity, evaluated the ecosystem service value, focused on estimating its ecological compensation standard, and analyzed the temporal and spatial distribution characteristics in Northeast China. I think this is a very valuable study, and the analytical method is regulated. The results have some important implications for the future refinement and advancement of cultivated land ecological compensation. However, some parts of the manuscript need to be revised. I recommend that a major revision is warranted. I ask that the authors to explain my concerns and specifically address each of my comments in their response.
My review below suggests some improvements.
1. Abstract: The abstract does not clearly write the innovation or contribution of the paper. And the sentence “The importance lies in the systematic assessment of the effectiveness of cultivated land ecological compensation, considering both ecological footprint and ecosystem service value” in abstract is not clear enough. The words “assessment of the effectiveness” are different from the words “assessment of cultivated land ecological compensation”.
2. Introduction: More space should be used to review whether the ecological footprint model and ESV can be combined and how to combine them.
3. Methods: The resolution of land use data is not clear.
Formula 10 is not correct enough.
4. Result: The clarity of figure 8 is not clear.
5. Discussion: This section that “4.2.3 Determine reasonable cultivated land compensation standards and promote positive interaction between stakeholders”
is not clearly written. It is suggested to further improve.
6. Conclusion: The conclusion is poor organization.
7. The format of references should be formatted according to LAND.
Comments on the Quality of English Language
Minor editing of English language required
Reviewer 3 Report
Comments and Suggestions for Authors
Dear authors, you started a good job; however I have some recommendations for its improvement:
Rows 107 – 115: there is duplicity of information: What does the research process include - it is described in rows 107-109 and then the rows 110-115 describe it again however, more detailed. Please rewrite this paragraph to unify the objectives of the paper.
Row 128 and others: please introduce the sum in the Chinese currency also in US dollar or Euro.
Row 130: …various crops….please introduce some examples which ones.
Table 1: please specify more detail equivalence factor and yield factor and GDP (GDP of China or only North China?). Moreover, GDP is not data unit – please specify it in a currency. Also the total population of the region in probably not expressed in person but number of persons. Please correct the data units in table 1.
Row 172: Please introduce a reference for the formula or is it your own?
Rows 176 – 178: please describe more details the variable Oi (it is average consumption in NC or a town in NC or in Chine as well?) and Pi – how was Pi calculated?
Row 186: Please clarify the part of formula (1-12%).
Row 217: you mean probably local GDP per capita and national GDP per capita, do not you? Please rewrite it in English.
Conclusions: Please mention in your conclusions answers to these questions:
· Why is this study unique?
· What are the shortcomings and uncertainties of this study?
· What did we scientific/research community learn out of it?
· Benefits for policymakers?
· Benefits for stakeholders?
· Future work?
Round 2
Reviewer 1 Report
Comments and Suggestions for Authors
This study lacks innovation and lacks a comprehensive understanding and resolution of previous research deficiencies. We hope that the author can further improve and revise it, rather than simply using others' materials as their own innovation. Figure 1. Lack of a north compass; CNY should be placed before the amount; The image on page 12 is missing a title.
Comments on the Quality of English LanguageNO
Reviewer 2 Report
Comments and Suggestions for Authors
1. The sentence “It provides a valuable reference for further adjustment and development of cultivated land ecological compensation” in line 16-17 should be deleted because it is highly similar to the last sentence in the abstract.
2. “Then” in line 18 should be deleted.
3.Formula 10 is not correct enough. “i” in Formula 8 is a subscript letter while not in Formula 10.
